# Altered Expression of GABAergic Markers in the Forebrain of Young and Adult *Engrailed-2* Knockout Mice

**DOI:** 10.3390/genes11040384

**Published:** 2020-04-01

**Authors:** Giovanni Provenzano, Angela Gilardoni, Marika Maggia, Mattia Pernigo, Paola Sgadò, Simona Casarosa, Yuri Bozzi

**Affiliations:** 1Department of Cellular, Computational and Integrative Biology (CIBIO), University of Trento, 38068 Rovereto, Italy; angela.gilardoni@yahoo.it (A.G.); marika.maggia@gmail.com (M.M.); perni.buz@gmail.com (M.P.); simona.casarosa@unitn.it (S.C.); 2Center for Mind/Brain Sciences (CIMeC), University of Trento, 38068 Rovereto, Italy; paola.sgado@unitn.it; 3C.N.R. Neuroscience Institute, 5624 Pisa, Italy

**Keywords:** interneuron, parvalbumin, somatostatin, GABA, receptor, hippocampus, somatosensory, autism, mouse, development

## Abstract

Impaired function of GABAergic interneurons, and the subsequent alteration of excitation/inhibition balance, is thought to contribute to autism spectrum disorders (ASD). Altered numbers of GABAergic interneurons and reduced expression of GABA receptors has been detected in the brain of ASD subjects and mouse models of ASD. We previously showed a reduced expression of GABAergic interneuron markers parvalbumin (PV) and somatostatin (SST) in the forebrain of adult mice lacking the *Engrailed2* gene (*En2^-/-^* mice). Here, we extended this analysis to postnatal day (P) 30 by using in situ hybridization, immunohistochemistry, and quantitative RT-PCR to study the expression of GABAergic interneuron markers in the hippocampus and somatosensory cortex of *En2^-/-^* and wild type (WT) mice. In addition, GABA receptor subunit mRNA expression was investigated by quantitative RT-PCR in the same brain regions of P30 and adult *En2^-/-^* and WT mice. As observed in adult animals, PV and SST expression was decreased in *En2^-/-^* forebrain of P30 mice. The expression of GABA receptor subunits (including the ASD-relevant *Gabrb3*) was also altered in young and adult *En2^-/-^* forebrain. Our results suggest that GABAergic neurotransmission deficits are already evident at P30, confirming that neurodevelopmental defects of GABAergic interneurons occur in the *En2* mouse model of ASD.

## 1. Introduction

Impaired function of GABAergic interneurons and subsequent alteration of excitation/inhibition balance in neural circuits has been implicated in neurodevelopmental disorders [1]. Forebrain GABAergic interneurons originate from the ganglionic eminences of the ventral forebrain, from where they migrate and differentiate during embryonic and early postnatal development [2]. Different classes of GABAergic interneurons are distinguished on the basis of their morphology, neurochemical content, intrinsic electrophysiological properties, and connectivity [3,4,5,6,7]. Defects in GABAergic neuron number and transmission have been hypothesized to contribute to autism spectrum disorders (ASD) [8,9,10]. Altered numbers of GABAergic interneurons were detected in the hippocampus of ASD subjects [11], and reduced GABA receptor density, number, and protein expression were also observed in the cerebellum and cortical areas of people with ASD [12]. Interneuron defects have also been reported in mouse models of ASD [10], suggesting that defects in the development and function of specific subsets of GABAergic interneurons might be responsible for behavioral impairment in ASD.

Genome-wide association studies identified *En2* as a candidate gene for ASD [13]; accordingly, mice lacking *En2* [14] display cerebellar defects resembling those reported in ASD subjects [15]. In addition, *En2^-/-^* mice present a series of pathological behaviors that are reminiscent of some features of ASD individuals, including reduced social interactions, defective spatial learning, and sensory processing deficits [16,17,18,19]. As frequently observed in ASD patients, *En2^-/-^* mice display an increased susceptibility to seizures [10,20] accompanied by the altered expression of ASD- and seizure-relevant genes [21]. We attributed this hyperexcitability to a reduced inhibitory innervation in forebrain areas of *En2^-/-^* mice, and were able to demonstrate that adult *En2^-/-^* mice do show a reduced number of parvalbumin (PV)-, somatostatin (SST)-, and neuropeptide Y (NPY)-positive interneurons in the hippocampus, somatosensory, and visual cortex [18,22,23].

Here, we investigated the expression of GABAergic markers parvalbumin (PV) and somatostatin (SST) and GABA receptor subunits in the hippocampus and somatosensory cortex of young *En2^-/-^* mice. Our results suggest that GABAergic neurotransmission deficits are already evident at an early postnatal age, confirming that neurodevelopmental defects of GABAergic interneurons occur in the *En2* mouse model of ASD.

## 2. Materials and Methods 

### 2.1. Animals

Experiments were approved by the Animal Welfare Committee of the University of Trento and Italian Ministry of Health (projects 949/2015-PR and 847/2018-PR), in accordance to the European Community Directive 2010/63/EU. Animals were housed in a 12 hr light/dark cycle with food and water available ad libitum, and all efforts were made to minimize suffering during experimental procedures. The generation and genetic background of *En2* mutants were previously described [14,22]. Wild type (WT) and *En2^-/-^* littermates used in this study were generated via heterozygous mating and genotyped as described [22]. A total of 44 age-matched young littermates (22 mice per genotype; 30 days old; weight = 20–25 g) of both sexes were used. Eight mice (4 per genotype) were used for histological morphometric analyses, 8 mice (4 per genotype) for in situ hybridization, 10 mice (5 per genotype) for immunohistochemistry, and 18 mice (9 per genotype) for quantitative RT-PCR experiments. An additional group of 8 age-matched adult littermates (4 mice per genotype; 3–5 months old; weight = 25–35 g) of both sexes were used for quantitative RT-PCR. Previous studies showed that similar group sizes are sufficient to obtain statistically significant results in histological, immunohistochemical, in situ hybridization, and RT-PCR studies [20,22]. All experiments were performed blind to genotype. Mice were assigned a numerical code by an operator who did not take part in the experiments, and codes were associated to genotypes only at the moment of data analysis. Experiments were performed on mice of both sexes, since previous studies did not reveal major significant differences between sexes at molecular, anatomical, and behavioral level [17,19,22].

### 2.2. Morphometric Analysis

Bright-field images of the hippocampus and somatosensory cortex were acquired at 10× primary magnification using a Zeiss M2 AxioImager microscope and merged using Adobe Photoshop software. Morphometric analysis of hippocampal and cortical layers was performed as described [22] on 2 to 4 cresyl-violet (Nissl) stained sections per animal, taken at the level of the dorsal hippocampus (*n* = 4 per genotype). Brain areas were identified according to the Allen Mouse Brain Atlas.

### 2.3. In Situ Hybridization

Brains from 4 WT and 4 *En2^-/-^* P30 mice were rapidly removed and frozen on dry ice. Twenty micrometer thick coronal sections were cut at the cryostat and fixed in 4% paraformaldehyde. Non-radioactive in situ hybridization was performed as described [20,22] using a mix containing glutamic acid decarboxylase (GAD) 65 (GenBank ID: M72422) and 67 (Genbank ID: NM_017007) digoxigenin-labeled riboprobes. Signal was detected by alkaline phosphatase-conjugated anti-digoxigenin antibody followed by NBT (p-nitroblue tetrazolium chloride)/BCIP (5-bromo-4-chloro-3-indolyl phosphate) alkaline phosphatase staining.

### 2.4. Quantitative RT-PCR

Total RNAs were extracted from dissected hippocampi and somatosensory cortices from 18 young (30 days old; 9 per genotype) and 8 adult (3–5 months old; 4 per genotype) mice, and retrotranscribed to cDNA according to published protocols [22,24]. qRT-PCR was performed in a C1000 Thermal Cycler (Bio-Rad), using the Kapa Probe Fast qPCR Master Mix (Resnova). The mouse housekeeping RNA for mitochondrial ribosomal protein L41 (mRPL41) was used as a standard for quantification. Primers (Eurofins Genomics) were designed on different exons to avoid amplification of genomic DNA (Table 1). Expression analyses were performed using the CFX3 Manager 3.0 (Bio-Rad) software [22,24]. Briefly, mean cycle threshold (Ct) values from triplicate experiments were calculated for each marker and L41, corrected for PCR efficiency and inter-run calibration. The expression level of each marker was then normalized to that of L41 (marker/L41 ratios) for WT and *En2^-/-^* mice.

### 2.5. Immunohistochemistry

Brains were fixed by transcardial perfusion (4% paraformaldehyde) followed by post-fixation (1 h at 4 °C), and vibratome sections (50 μm thickness) were prepared. Serial sections at the level of the dorsal hippocampus were incubated overnight with the following antibodies; anti-parvalbumin (PV) mouse monoclonal (P3088, Sigma-Aldrich, USA; 1:2000 dilution); anti-somatostatin (SOM) rabbit polyclonal (T-4102, Bachem, UK; 1:2000 dilution); anti-GABA_A_ receptor α1 subunit (GABRA1) rabbit polyclonal (ab33299, Abcam, 1:1000 dilution); anti-GABA_A_ receptor α3/GABRA3) rabbit polyclonal (AB5594, Millipore, 1:1000 dilution); and anti-GABA_A_ receptor α5 (GABRA5) rabbit polyclonal (ab 10098, Abcam, 1:1000 dilution). Biotin-conjugated secondary antibody and streptavidin conjugated to appropriate fluorophores (AlexaFluor 488/594, Invitrogen Life Technologies, USA) were used to reveal the signal.

### 2.6. Cell Counts

Cell counts were performed on acquired images of the hilus and somatosensory cortex, using the ImageJ software (imagej.nih.gov/ij/index.html), as described [22]. Brain areas were identified according to the Allen Mouse Brain Atlas (http://mouse.brain-map.org). For in situ hybridization, counts of GAD65/67 mRNA labeled cells were performed on 3 sections per animal (*n* = 4 per genotype) at the level of the dorsal hippocampus/somatosensory cortex. For each section, bright-field images were acquired at 10× primary magnification using a Zeiss M2 AxioImager microscope and merged using the Zeiss software. For immunohistochemistry, 3 to 4 immunolabeled sections were analyzed per animal (5 mice per genotype). For each section, images were acquired at 20× primary magnification using a Zeiss Axio Observer z1, using the MosaiX and Z-Stack modules of the Zeiss AxioVision software (v4.3.1). The same counting procedure was followed for both in situ hybridization (GAD65/67) and immunohistochemistry (PV/SST) images. To count positive cells in the hilus, the total hilar area was measured excluding the granule cell layer, and did not differ between WT and *En2^-/-^* mice. Cell densities were plotted as the number of positive cells/0.1 mm^2^. For the somatosensory cortex, cell densities were separately counted in superficial (II-III) and deep (V-VI) layers using at least three counting frames (200 × 600 μm) per section, and plotted as the number of positive cells/counting area. All counts were performed by two independent experimenters blind of genotypes.

### 2.7. Statistical Analyses

Statistical analyses were performed by Prism 6 (GraphPad) software. Student’s t-test or two-way ANOVA followed by appropriate post hoc test (as indicated) were used, with statistical significance level set at *p* < 0.05.

## 3. Results

Previous studies from our laboratory showed the reduced expression of GABAergic markers in the hippocampus and somatosensory cortex of adult (3–5 months old) *En2^-/-^* mice, as compared with age-matched controls [20,22]. Here, we extended this analysis to an earlier stage of postnatal development (postnatal day 30, P30).

### 3.1. Normal Layering of Hippocampus and Somatosensory Cortex in Young En2 Mutant Mice

We first performed a morphometric analysis on dorsal hippocampal and somatosensory cortex sections stained with cresyl violet to investigate the layering of these structures in P30 WT and *En2^-/-^* mice. A normal anatomical layer structure was detected in both WT and *En2^-/-^* mice in the two brain areas analyzed. Total hippocampal thickness did not differ between the two genotypes (WT: 1091 ± 21 μm, *n* = 17 sections from 4 mice; *En2^-/-^*: 1090 ± 18 μm, *n* = 11 sections from 4 mice). Similarly, the total thickness of the somatosensory cortex (WT, 1253.54 ± 27 μm, *n* = 15 sections from 4 mice; *En2^-/-^*, 1260.10 ± 34 μm, *n* = 19 sections from 4 mice) did not differ between WT and *En2^-/-^* mice (*p* > 0.05, Student’s *t*-test). No difference was detected in hippocampal (Figure 1A) and cortical (Figure 1B) layer thickness between the two genotypes (*p* > 0.05, two-way ANOVA followed by Holm–Sidak test).

We next investigated the expression of GABAergic interneuron markers in the hippocampus of P30 WT and *En2^-/-^* mice. GABAergic interneurons are characterized by the expression of specific markers [3,4,5,6,7]. The total number of GABAergic neurons, estimated by in situ hybridization for GAD65/67 mRNA, was unchanged in the hilus and somatosensory cortex (Figure 2A,B) of *En2^-/-^* mice, as compared to WT controls. 

We previously reported a significantly reduced expression of parvalbumin (PV) and somatostatin (SST) GABAergic interneuron markers in both hippocampus and somatosensory cortex of adult *En2^-/-^* mice, as compared with age-matched WT animals [22]. Immunohistochemistry experiments performed on brain sections from P30 mice revealed a comparable number of PV-positive cells in both hilus and somatosensory cortex (layers II-III and V-VI) of P30 *En2^-/-^* mice (Figure 3A,C; *p* > 0.05, two-way ANOVA followed by Holm–Sidak test); similarly, the number of hilar SST-positive interneurons did not differ between the two genotypes (Figure 3B,E; *p* > 0.05, two-way ANOVA followed by Holm–Sidak test). A statistically significant reduction of SST interneurons was instead detected in layers II–III and V–VI of the *En2^-/-^* somatosensory cortex (−25%), as compared to age-matched WT controls (Figure 3B,E; *** *p* < 0.001, two-way ANOVA followed by Holm–Sidak test). Quantitative RT-PCR on total RNAs extracted from WT and *En2^-/-^* hippocampi and somatosensory cortices showed contrasting data with respect to immunohistochemistry experiments. PV mRNA levels were significantly reduced in the *En2^-/-^* hippocampus (−55%) and somatosensory cortex (−27%) as compared with WT controls (Figure 3D; * *p* < 0.05, *** *p* < 0.001, two-way ANOVA followed by Holm–Sidak test). Conversely, SST mRNA levels did not differ between the two genotypes in both brain areas analyzed (Figure 3F; *p* > 0.05, two-way ANOVA followed by Holm–Sidak test).

Table 2 summarizes in situ hybridization, immunohistochemistry, and RT-PCR data on interneuron marker expression in the hippocampus and somatosensory cortex of P30 WT and *En2^-/-^* mice. Taken together, these data indicate that, as already observed in adult animals, the *En2* null mutation results in the altered expression of PV and SST interneuron markers in the hippocampus and somatosensory cortex of young mice.

### 3.2. Altered Expression of GABA Receptor Subunits in Young and Adult En2 Mutant Mice

Several human and animal studies indicate that altered GABA receptor subunit expression contributes to impaired GABAergic neurotransmission in ASD [10,25,26]. In order to identify the potential mechanisms of GABA receptor dysfunction in our mouse model of ASD, we investigated the expression of GABA receptor subunits in the hippocampus and somatosensory cortex of P30 and adult WT and *En2^-/-^* mice by using quantitative RT-PCR. Specifically, we analyzed the mRNA of several different subunits that mainly contribute to ionotropic GABA_A_ and metabotropic GABA_B_ receptors [27,28,29]: GABA_A_ receptor subunits α1, α2, α3, α5 (*Gabra1-5*), β2, β3 (*Gabrb2-3*), γ2 (*Gabrg2*), and GABA_B_ receptor subunit 2 (*Gabbr2*).

At P30, *Gabra5*, *Gabrab3*, and *Gabrg2* mRNAs were upregulated in the *En2^-/-^* hippocampus, as compared to age-matched controls (+30%, +30%, and +43%, respectively; ** *p* < 0.01, Student’s t-test, Figure 4A), while all other subunit mRNAs were unchanged (*p* > 0.05, Student’s *t*-test, Figure 4A). In the somatosensory cortex, only *Gabrab3* and *Gabbr2* mRNAs significantly differed between the two genotypes (respectively, +30% and −26% in *En2^-/-^* as compared with WT mice; ** *p* < 0.01, Student’s *t*-test, Figure 4A). The expression profile of GABA receptor subunits markedly differed during adulthood. In the somatosensory cortex, only *Gabra1* mRNA was significantly lower in *En2^-/-^* as compared with WT mice (−17%; ** *p* < 0.01, Student’s *t*-test, Figure 4B), while all other subunit mRNAs were comparably expressed in both genotypes (*p* > 0.05, Student’s *t*-test, Figure 4B). Finally, in the hippocampus, *Gabra3*, *Gabrb2*, *Gabrb3*, and *Gabbr2* mRNAs significantly differed between the two genotypes (respectively, +10%, −33%, −20%, and −10% in *En2^-/-^* as compared with WT mice; * *p* < 0.05, ** *p* < 0.01, Student’s *t*-test, Figure 4B), while the other subunit mRNAs remained unchanged (*p* > 0.05, Student’s *t*-test, Figure 4B).

Table 3 summarizes the expression profile of GABA receptor subunit mRNAs in the hippocampus and somatosensory cortex of P30 and adult WT and *En2^-/-^* mice. Taken together, these data show that mice lacking *En2* display an altered expression of several GABA_A_ and GABA_B_ receptor subunits in the hippocampus and somatosensory cortex of young and adult mice.

We previously showed that the protein levels of the GABA_A_ receptor subunit β3 were decreased in the adult *En2^-/-^* hippocampus, thus confirming mRNA expression data [30]. Here, we investigated the protein expression of GABA_A_ receptor subunits α1, α3, and α5 in the hippocampus of P30 WT and *En2^-/-^* mice. Immunohistochemistry experiments showed a slight increase of GABRA1 and GABRA5 staining in the *En2^-/-^* dentate gyrus, as compared to WT controls; conversely, GABRA3 staining was comparable in both genotypes (Figure 5).

## 4. Discussion

### 4.1. Brief Summary of Results

In this study, we showed that young (P30) *En2^-/-^* mice display an altered expression of interneuron markers, namely, a reduced expression in PV mRNA in the hippocampus and somatosensory cortex and a reduced number of SST-positive cells in the somatosensory cortex. The expression of GABA receptor subunit mRNAs was also altered in these brain areas of young and adult *En2* mutants. The GABA_A_ receptor β3 subunit (*Gabrb3*) was the most severely affected, showing a significant upregulation in both areas at P30, and a downregulation in the adult hippocampus. Among the other subunits, α5 showed a consistent increase of both mRNA and protein levels in the hippocampus of young *En2* mutant mice.

### 4.2. Altered Expression of GABAergic Interneuron Markers in Young En2 Mutant Mice

Previous studies from our laboratory showed a reduced expression of GABAergic interneuron markers in the forebrain of adult *En2^-/-^* mice. The present results, obtained from P30 mice, expand this notion and confirm that *En2* contributes to regulate the function of forebrain GABAergic interneurons already at early postnatal ages. As previously observed in adult mice [22], the loss of *En2* resulted in the reduced expression of PV and SST in the hippocampus and somatosensory cortex of young P30 mice, without affecting the total number of GAD-positive interneurons, nor the layering of these brain areas. More specifically, a 25% reduction in SST-positive cells was consistently detected in layers II–III of P30 (this study) and adult [22] *En2^-/-^* mice. This data confirms that the *En2* mutation impacts the expression of specific GABAergic interneuron markers during postnatal development. This selective effect is likely due to altered embryonic development of GABAergic interneurons in *En2* mutant brains. Indeed, neural stem cells derived from embryonic *En2^-/-^* basal forebrain are not capable to properly differentiate into mature neurons, and do show a reduced expression of PV and SST GABAergic markers during in vitro differentiation [31]. However, the expression profile of PV and SST in *En2* mutant brains significantly differed between P30 and adult mice. Although both PV mRNA and cell number were shown to be reduced in the adult *En2^-/-^* hippocampus and somatosensory cortex [22], only PV mRNA (but not cell number) was decreased in the same brain areas of young *En2* mutants. Similarly, SST mRNA downregulation was detected only in the adult [22] but not P30 hippocampus. Interestingly, the downregulation of PV expression in the absence of a PV cell loss has been observed in other mouse models of ASD such as *Cntnap2^-/-^* and *Shank3b^-/-^* mice, as well as mice exposed in utero to valproic acid. In all these models, no loss of PV neurons was detected in ASD-relevant brain regions including the striatum, whereas PV protein levels were decreased in this area [32,33,34]. The authors of these studies proposed that the observed downregulation of PV in the striatum of these mouse models might represent a homeostatic mechanism to counteract reduced inhibition, thus to maintain a physiological excitation/inhibition balance [34,35]. A similar mechanism might therefore be in place in the hippocampus and somatosensory cortex of young *En2* mutants, which showed reduced PV mRNA (but not cell number) in these same brain areas. Taken together, these results strengthen the notion that specific markers of hippocampal and cortical interneurons are affected in mouse models of ASD during postnatal development.

### 4.3. Altered Expression of GABA Receptor Subunits in Young and Adult En2 Mutant Mice

Abnormal GABAergic inhibition has been proposed as a pathogenic mechanism of multiple neurodevelopmental disorders, including ASD [10]. As mentioned above, GABAergic interneurons are clearly affected in human ASD, as shown by post-mortem studies performed on hippocampal tissues from ASD patients compared to control samples from typically-developing subjects [11]. Similar results have been obtained in mouse models of ASD [10]. GABA receptor dysfunction has also been detected in ASD subjects and mouse models. For example, reduced ionotropic GABA_A_ receptor subunits and benzodiazepine binding were detected in the cerebral cortex of ASD individuals [36,37,38,39,40]. Among the several different GABA_A_ receptors, the α5 subtype gained the attention of ASD researchers. Single nucleotide polymorphisms (SNPs) across the GABRA5 and GABRB3 genes (respectively, coding for the α5 and β3 subunits of the GABA_A_ α5 receptor) are associated with ASD [41,42]. A reduced density of GABA_A_ α5 receptors was initially detected in the brain of ASD patients [43], a result that was not replicated in further studies [44]. Mice lacking the β3 subunit of the GABA_A_ receptor (*Gabrb3^-/-^* mice) show ASD-relevant traits such as reduced sociability [45] and abnormal response to tactile stimulation [46], thus strengthening the importance of the GABA_A_ α5 receptor subtype in ASD pathogenesis.

In the present study, we analyzed the expression of GABA_A_ and GABA_B_ receptor subunit mRNAs in the hippocampus and somatosensory cortex of P30 and adult *En2^-/-^* and WT mice. With the exception of *Gabra2*, all other subunit mRNAs were deregulated in at least one structure and at one age of the two analyzed (Table 3). Interestingly, the most consistently deregulated was *Gabrb3* mRNA, which codes for the β3 subunit of the GABA_A_ receptor. *Gabrb3* mRNA was upregulated in the *En2^-/-^* hippocampus and somatosensory cortex at P30, while in adult *En2^-/-^* mice, it showed a significant downregulation only in the hippocampus. These results indicate that *Gabrb3* mRNA in the hippocampus undergoes a time-dependent regulation in *En2^-/-^* mice; in addition, the *Gabrb3* mRNA upregulation has never been observed in other mouse models of ASD. More importantly, the results obtained in adult mice are in agreement with our previous study that showed a marked downregulation of the FMRP and GABRB3 protein in the adult *En2^-/-^* hippocampus [30]. In accordance with these results, reduced levels of FMRP and GABRB3 proteins were detected in the cerebellum of adult ASD subjects [38]. Interestingly, mice lacking FMRP (*Fmr1^-/-^* mice), which share anatomical and functional abnormalities with *En2* mutants [19,47,48], also show a decreased expression of GABA_A_ receptor subunit mRNAs and proteins [49,50]. Thus, FMRP-dependent downregulation of GABA_A_ β3 subunit expression might contribute to alter GABAergic transmission in *En2* mutant mice. Conversely, the α5 subunit showed a consistent increase in both mRNA and protein levels in the hippocampus of young *En2* mutant mice. Increased levels of *Gabra5* mRNA have been reported in the cerebellum of ASD subjects, whereas a reduced expression was detected in neocortical areas [37]. In keeping with these findings, GABA_A_ receptors containing the α5 subunit were initially proposed as a potential pharmacological target in ASD [43], although these results were not corroborated by subsequent studies [44]. Regarding the expression of the GABA_A_ α1 subunit, we reported discordant results between its mRNA and protein levels in the hippocampus of P30 *En2* mutants (Figure 4 and Figure 5). Different mechanisms, including the downregulation of microRNAs inhibiting GABRA1 protein translation, or post-translational modifications altering its trafficking, might be involved. Interestingly, studies performed on post-mortem brain samples from ASD subjects also reported discordant mRNA and protein expression profiles of GABA receptor subunits [37].

Finally, in agreement with previous studies on ASD mouse models [15,50] and human subjects [37], GABA receptor subunits expression varied across different brain regions in *En2* mutants, and the detected changes between WT and *En2^-/-^* mice were relatively small. Nevertheless, the entity of the expression changes detected in *En2* mutants (ranging from 10% to 43%) was comparable to what observed in *Fmr1* knockout mice [50], which definitely show compromised GABAergic transmission [51]. We therefore speculate that these alterations might be enough to modify GABA signaling, thus substantiating a pathological readout also in *En2^-/-^* mice.

## 5. Conclusions

Previous studies from our laboratory showed that the expression of several markers of GABAergic synaptic function (receptors, structural proteins, signaling molecules, etc.) are deregulated in the forebrain of *En2^-/-^* mice compared to WT littermates [18,22,23]. The results reported in the present study confirm these observations, strengthening the notion that GABAergic dysfunction is a common feature of several mouse models of ASD. Most importantly, our results indicate that GABAergic signaling dysfunction might contribute the complex behavioral phenotype of *En2* mutants since early postnatal ages. Together with other studies addressing early postnatal deficits of GABA transmission in ASD mouse models [49,52,53,54], our findings might guide future research towards the identification of precocious markers of GABAergic dysfunction in ASD.

## Figures and Tables

**Figure 1 genes-11-00384-f001:**
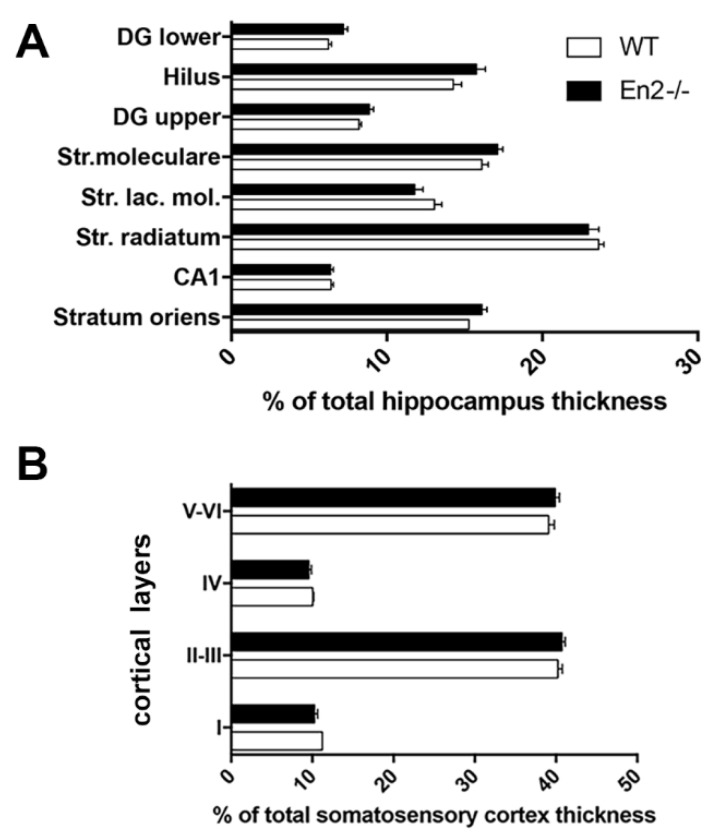
Altered expression of GABAergic interneuron markers in young *En2* mutant mice. Normal layering of the young *En2^-/-^* hippocampus and somatosensory cortex. Morphometric analysis of hippocampal (**A**) and somatosensory cortex (**B**) layers in young (P30) WT (wild type) and *En2^-/-^* mice (*n* = 4 mice per genotype). Layer thickness is plotted as % of total thickness of the dorsal hippocampus (**A**) and somatosensory cortex (**B**). Genotypes and layers are as indicated; cortical layers are indicated in roman numbers. Abbreviations: DG lower: lower blade of the dentate gyrus; DG upper: upper blade of the dentate gyrus; Str.: stratum; Str. lac. mol.: stratum lacunosum moleculare.

**Figure 2 genes-11-00384-f002:**
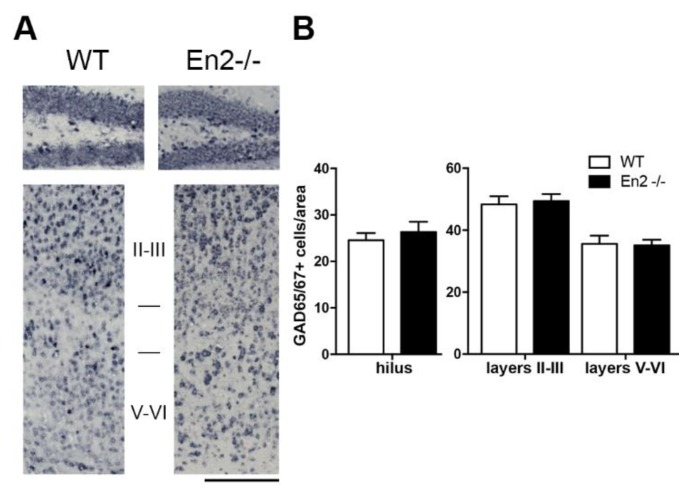
Unaltered number of GABAergic interneurons in the young *En2^-/-^* hippocampus and somatosensory cortex. (**A**) Representative photomicrographs of in situ hybridizations showing GAD65/67 mRNA-positive neurons in the hilus (top) and somatosensory cortex (bottom) of P30 WT and *En2^-/-^* mice. Scale bar: 600 μm. Cortical layers are indicated in Roman numbers. (**B**) GABAergic (GAD65/67 mRNA-positive) interneuron cell counts from in situ hybridization experiments. Values are plotted as the mean number (± s.e.m) of positive cells per area (0.1 mm^2^ for the hilus and 0.12 mm^2^ for the somatosensory cortex) from 4 mice per genotype. Genotypes are as indicated. Abbreviations: hippo: hippocampus; ss ctx: somatosensory cortex.

**Figure 3 genes-11-00384-f003:**
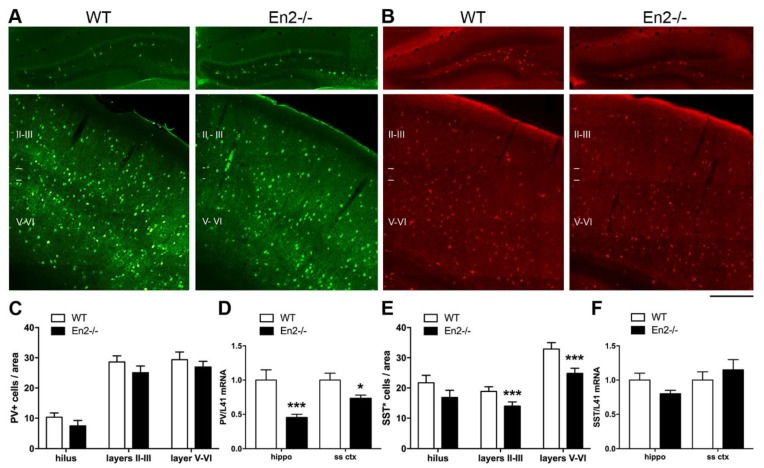
Reduced expression of parvalbumin (PV) and somatostatin (SST) interneuron markers in the young *En2^-/-^* hippocampus and somatosensory cortex. (**A**,**B**) Representative immunostainings of PV ((**A**), green) and SST ((**B**), red) interneurons in the hilus (top) and somatosensory cortex (bottom) of P30 WT and *En2^-/-^* mice. Cortical layers are indicated in Roman numbers. Scale bar: 400 μm. (**C**) PV cell counts from immunostaining experiments, plotted as the mean number (± s.e.m) of positive cells per area (0.1 mm^2^ for the hilus and 0.12 mm^2^ for the somatosensory cortex) on 5 mice per genotype. (**D**) PV mRNA expression in the hippocampus and somatosensory cortex of P30 WT and *En2^-/-^* mice. Values are plotted as the mean ± s.e.m of PV/L41 comparative quantification ratio (* *p* < 0.05; *** *p* < 0.001; Student’s t-test, WT vs. *En2^-/-^*, 4 WT and 6 *En2^-/-^* mice). (**E**) SST cell counts from immunostaining experiments, plotted as in C (*** *p* < 0.001, two-way ANOVA followed by Sidak’s multiple comparisons test, WT vs. *En2^-/-^*, 5 mice per genotype). (**F**) SST mRNA expression in the hippocampus and somatosensory cortex of P30 WT and *En2^-/-^* mice (n = 4 WT and 6 *En2^-/-^*). Values are plotted as in (**D**). Genotypes are as indicated. Abbreviations: PV, parvalbumin; SST, somatostatin.

**Figure 4 genes-11-00384-f004:**
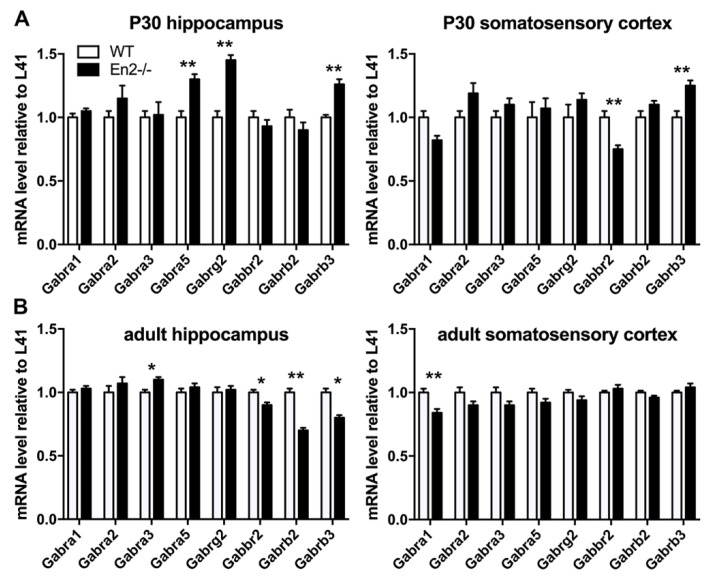
mRNA expression of GABA receptor subunits in the forebrain of young and adult WT and *En2^-/-^* mice. (**A**,**B**) Relative mRNA expression level of GABA receptor subunits, as obtained by quantitative RT-PCR performed on hippocampus and somatosensory cortex of P30 (**A**) and adult (**B**) WT and *En2^-/-^* mice. Values are expressed as the mean ± s.e.m of each subunit/L41 comparative quantification ratios (4 mice per age and genotype; * *p* < 0.05, ** *p* < 0.01, Student’s *t*-test, WT vs. *En2^-/-^*). Brain areas, ages, and genotypes are as indicated. Abbreviations are as in the text.

**Figure 5 genes-11-00384-f005:**
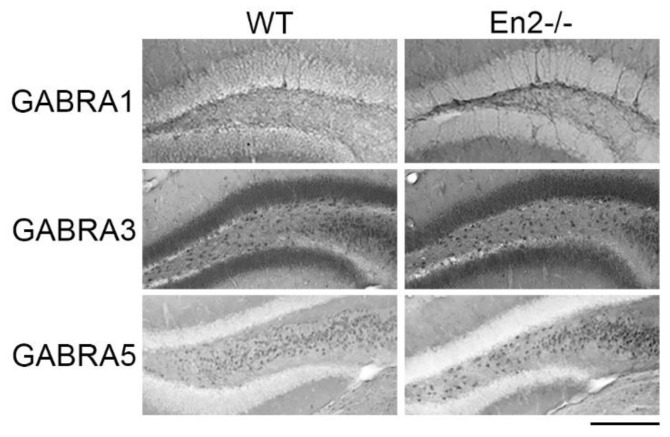
Expression of GABA_A_ receptor subunits in dentate gyrus of young WT and *En2^-/-^* mice. Pictures show representative immunostainings of GABRA1, GABRA3, and GABRA5 subunits in the dentate gyrus of P30 WT and *En2^-/-^* mice. Scale bar: 300 μm. Genotypes are as indicated. Abbreviations are as in the text.

**Table 1 genes-11-00384-t001:** Primers used for quantitative RT-PCR experiments.

Gene Name	Genbank #	Forward Primer (5′–3′)	Reverse Primer (5′–3′)
*L41*	NM_001031808.2	GGTTCTCCCTTTCTCCCTTG	GCACCCCGACTCTTAGTGAA
*PV*	NM_013645	TGCTCATCCAAGTTGCAGG	GCCACTTTTGTCTTTGTCCAG
*SOM*	NM_009215	AGGACGAGATGAGGCTGG	CAGGAGTTAAGGAAGAGATATGGG
*Gabra1*	NM_010250	CTCTCCCACACTTTTCTCCC	CCGACAGTGTGCTCAGAATG
*Gabra2*	NM_008066.4	AGATTCAAAGCCACTGGAGG	CCAGCACCAACCTGACTG
*Gabra3*	NM_001357816.1	CAGACTGAGATAGGGACTAGGAG	AGACAGCAACTTGAAGAGACC
*Gabra5*	NM_176942.4	CCCTATCCCAACACCTGAAC	AATGTTCAAAGGGTTCTGCC
*Gabrg2*	NM_008073.4	CACCGGGCATGAATGTG	GGATGGTACACGCAGAGATG
*Gabrb2*	NM_001347314	TCAGAGGATGACTTTGCTA	GCACACAATAATGTTTACTAT
*Gabrb3*	NM_008071.3	GAGGTCTTCACAAGCTCAAAATC	AGGCAGGGTAATATTTCACTCAG
*Gabbr2*	NM_031802.1	ACATGCAAAGACCCCATAGAG	TCGTGAGAGTAAGACCGTCG

**Table 2 genes-11-00384-t002:** Expression of interneuron markers in the hippocampus and somatosensory cortex of *En2^-/-^* P30 mice.

Interneuron Marker	Hippocampus		Somatosensory Cortex		
	mRNA	Cell Counts *(hilus)*	mRNA	Cell Counts *(layers II–III)*	Cell Counts *(layers V–VI)*
GAD	not tested	no difference	not tested	no difference	no difference
PV	−55%	no difference	−27%	no difference	no difference
SST	no difference	no difference	no difference	−25%	−25%

Data are presented as the mean percentage of the reduction of interneuron marker mRNA and positive cells detected in *En2^-/-^* mice, as compared to WT littermates. See text and figures for experimental details. Abbreviations as in the text.

**Table 3 genes-11-00384-t003:** mRNA expression of GABA receptor subunits in the hippocampus and somatosensory cortex of *En2^-/-^* P30 and adult mice.

Receptor Subunit	P30		Adult	
	Hippocampus	Somatosensory Cortex	Hippocampus	Somatosensory Cortex)
*Gabra1*	no difference	no difference	no difference	−17%
*Gabra2*	no difference	no difference	no difference	no difference
*Gabra3*	no difference	no difference	+10%	no difference
*Gabra5*	+30%	no difference	no difference	no difference
*Gabrb2*	no difference	no difference	−33%	no difference
*Gabrb3*	+30%	+30%	−20%	no difference
*Gabrg2*	+43%	no difference	no difference	no difference
*Gabbr2*	no difference	−26%	−10%	no difference

Data are presented as the mean percentage of the reduction of GABA receptor subunit mRNAs in *En2^-/-^* mice, as compared with WT littermates. See text and figures for experimental details. Abbreviations as in the text.

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
