# Peer review of "Altered Expression of GABAergic Markers in the Forebrain of Young and Adult Engrailed-2 Knockout Mice"

_genes, 2020, doi:10.3390/genes11040384_

Round 1
Reviewer 1 Report
In the present study, the author looked at the altered expression of GABAergic markers in the forebrain of young and adult Engrailed-2 KO mice. Although they did not observe any overall difference in the number of interneurons in the brain, they found an altered level of certain GABA markers. While the findings are interesting and are an extension of previous work from the lab, the manuscripts lack novelty in terms of research question and how the findings can contribute to future research:
Comments:
1) While qRT-PCR is a widely used method to look at gene expression, it is hard to compare its data to cell numbers and other immunohistochemistry data since one looks ar mRNA and other at proteins. The authors are strongly recommended to perform some western blots at least for the significantly altered subunits to strengthen their findings, especially because in most cases, the difference in mRNA level is not particularly robust.
2) GABA receptor subunit expression varies across different brain regions and that may be included in the discussion and how the results are affected because of the same.
3) The images in Figure 3 lacks a general cell/ nucleus marker such as DAPi which makes it harder to interpret the change in specific cell number compared to total cell number.
4) In the discussion, only increase in GABArb3 has been discussed while the other changes and how it can functionally affect the mice phenotype overall has not been discussed.
5) Figure 3A: the background in WT seems much higher than EN2-/- and a using a more appropriate representative images is recommended.
Author Response
General comments
“The manuscript lacks novelty in terms of research question and how the findings can contribute to future research”.
We modified the Conclusion paragraph of the Discussion (page 16) to address this point as follows: “our results indicate that GABAergic signaling dysfunction might contribute the complex behavioural phenotype of En2 mutants since early postnatal ages. Together with other studies addressing early postnatal deficits of GABA transmission in ASD mouse models [49,53,54], our findings might guide future research towards the identification of precocious markers of GABAergic dysfunction in ASD”.
Major points
1) “The authors are strongly recommended to perform some western blots at least for the significantly altered subunits to strengthen their findings, especially because in most cases, the difference in mRNA level is not particularly robust”.
We agree with the Reviewer that qRT-PCR data should be confirmed by quantitative western blot analysis. However, due to the current coronavirus emergency, we are not in the condition (nor we can foresee when we will be able) to perform the requested experiments. To partially address the Reviewer comment, we now provide representative images of immunohistochemistry experiments for GABAA receptor subunits a1, a 3, and a 5, on P30 hippocampi from WT and mutant mice (new Figure 5, Results page 12). These experiments confirm the mRNA expression data already obtained for a 3 and a 5 subunits. Their relevance is also discussed in the Discussion (page 15).
2) “GABA receptor subunit expression varies across different brain regions and that may be included in the discussion and how the results are affected because of the same”.
We acknowledged and discussed this issue in the Discussion (page 15).
3) “The images in Figure 3 lacks a general cell/nucleus marker such as DAPI which makes it harder to interpret the change in specific cell number compared to total cell number”.
As already discussed for point 1, we cannot address this point experimentally. However, data presented in Figure 2 clearly indicate that the total number of GABAergic (GAD positive) neurons is comparable in WT and mutant mice. This is reported in page 13 (Discussion paragraph “Altered expression of GABAergic interneuron markers in young En2 mutant mice”), where we state that “…the loss of En2 resulted in the reduced expression of PV and SST in the hippocampus and somatosensory cortex of young P30 mice, without affecting the total number of GAD-positive interneurons…”.
4) “In the discussion, only increase in GABArb3 has been discussed while the other changes and how it can functionally affect the mice phenotype overall has not been discussed”.
We now discuss (Discussion, page 15) also data regarding GABAA receptor subunits a1, and a 5.
5) “Figure 3A: the background in WT seems much higher than EN2-/- and a using a more appropriate representative images is recommended”.
We have modified Figure 3A (parvalbumin immunohistochemistry) as requested.
Reviewer 2 Report
Overall the paper is well written and the experiments logic and rational. Also the techniques adopted are suitable for the aim of the paper and replicates robust.
My main concerns are the lack of novelty and the significance of the results.
This paper is very similar to a previous work, but now the authors analyse gene expression changes in the same mutant but at a different time-point. The other main criticism is that gene expression changes observed are very minimal and very often qPCR results are contrasting with IHC ones.
Detailed comments below:
Major
3.2 results section
- Why a mixed probe for GAD65 and GAD67 isoforms was used for ISH and for qPCR only GAD67 was detected? For clarity I think that both qPCR should be run.
- How are the contrasting data between qPCR and IHC interpreted?
- The changes in GABA receptor genes are very minimal; do the authors think that these changes, mainly detected with qPCR, are enough to substantiate a pathological readout?
Minor
-Some typos or grammar mistakes to be edited through the text.
Examples:
Line 36 full stop is missing after ‘development’
Line 43 change ‘was’ with ‘were’
- In the methods section it is stated that ISH for GABAergic interneuron markers was done on adult brains, but in the results section it is reported to be p30 mice. Please reconcile.
-Line 98 please spell out the substrate used for alkaline phosphatase staining (ISH experiments)
Author Response
Reviewer 2
General comments
“My main concerns are the lack of novelty and the significance of the results. This paper is very similar to a previous work, but now the authors analyse gene expression changes in the same mutant but at a different time-point”.
As explained in our response to Reviewer 1, we modified the Conclusion paragraph of the Discussion (page 16) to address this point.
“The other main criticism is that gene expression changes observed are very minimal and very often qPCR results are contrasting with IHC ones”.
We agree with the reviewer that qRT-PCR and immunohistochemistry data may be discordant, as it is the case for our data on the expression of interneuron markers reported in Figure 3 and Table 2. We provide an explanation for this discordance in the Discussion, page 14. As for GABA receptor subunit expression, we now provide immunohistochemistry data for a1, a 3, and a 5 subunits, and discuss it with respect to mRNA expression data (page 15). Finally, as regarding the minimal expression changes detected by qRT-PCR, we now discuss (pages 15-16) that “the entity of the expression changes detected in En2 mutants (ranging from 10% to 43%) was comparable to what observed in Fmr1 knockout mice [50], which definitely show compromised GABAergic transmission [52]. We therefore speculate that these alterations might be enough to modify GABA signaling thus substantiating a pathological readout also in En2-/- mice”.
Major points
1) “Why a mixed probe for GAD65 and GAD67 isoforms was used for ISH and for qPCR only GAD67 was detected? For clarity I think that both qPCR should be run”.
We agree with the Reviewer that GAD expression data, as previously presented, were misleading. Since we are not in the condition to perform additional RT-PCR experiment at this time, we decided to remove qRT-PCR data for GAD67 mRNA (Figure 2C), which did not add much to in situ hybridization data. We now provide a new version of Figure 2 only showing in situ hybridization data for GAD65/67.
2) How are the contrasting data between qPCR and IHC interpreted?
As explained above, we addressed this issue in the Discussion at page 14 (for PV/SST) and 15 (for GABA receptor subunits.
3) The changes in GABA receptor genes are very minimal; do the authors think that these changes, mainly detected with qPCR, are enough to substantiate a pathological readout?
To address this issue, we added immunohistochemistry data for a1, a 3, and a 5 subunits (Figure 5). As regarding the minimal expression changes detected by qRT-PCR, we now discuss (pages 15-16) that the entity of these changes was comparable to that observed in another ASD mouse model (Fmr1 knockout mice) that show important anatomical and functional defects similar to those observed in En2 mutants.
Minor points
“Line 36 full stop is missing after ‘development’”. This typo has been corrected.
“Line 43 change ‘was’ with ‘were”’ This typo has been corrected.
“In the methods section it is stated that ISH for GABAergic interneuron markers was done on adult brains, but in the results section it is reported to be p30 mice”. This typo has been corrected.
“Line 98 please spell out the substrate used for alkaline phosphatase staining (ISH experiments)”. We now specify that we used NBT/BCIP staining.
Round 2
Reviewer 1 Report
The paper can be accepted in its current form
Reviewer 2 Report
The authors had improved significantly the quality of the manuscript by taking into account my comments.